# Sound Event Detection in Underground Parking Garage Using Convolutional Neural Network

**Giuseppe Ciaburro**

Department of Architecture and Industrial Design, University of Campania Luigi Vanvitelli, 81031 Aversa, Italy; giuseppe.ciaburro@unicampania.it

**Abstract:** Parking is a crucial element in urban mobility management. The availability of parking areas makes it easier to use a service, determining its success. Proper parking management allows economic operators located nearby to increase their business revenue. Underground parking areas during off-peak hours are uncrowded places, where user safety is guaranteed by company overseers. Due to the large size, ensuring adequate surveillance would require many operators to increase the costs of parking fees. To reduce costs, video surveillance systems are used, in which an operator monitors many areas. However, some activities are beyond the control of this technology. In this work, a procedure to identify sound events in an underground garage is developed. The aim of the work is to detect sounds identifying dangerous situations and to activate an automatic alert that draws the attention of surveillance in that area. To do this, the sounds of a parking sector were detected with the use of sound sensors. These sounds were analyzed by a sound detector based on convolutional neural networks. The procedure returned high accuracy in identifying a car crash in an underground parking area.

**Keywords:** sound classification; convolutional neural networks; audio event detection; acoustic measurements; acoustic features

## 1. Introduction

The appearance of new situations of social alarm linked to a lack of human resources has led to the progressive use of technologies to which some of the safety prerogatives are delegated. Technological development has made new tools available to guarantee high levels of safety for people while socializing, who require ever higher standards in reducing the risk associated with their own safety. In different contexts, large amounts of data are collected, which can be used to develop applications and services that can simplify our daily lives. In this way, the collected data can be processed to extract information on social phenomena and guide the processes of developing new and more incisive policies. The main component of road traffic is represented by private cars. Each movement of a vehicle always takes place from a departure to a destination, and, in the vicinity of both, the vehicle will have to remain stationary for some time: on average, more than 90% of a car's life is spent in parking conditions. Parking, especially in large cities, is a factor which has a great impact on the flow of traffic and on the level of stress of people in finding a parking space. As the number of vehicles increases, the problem of parking management increases [1]. Parking lots have become essential near large public buildings such as hospitals, shopping malls, hotels, and airports. The absence of parking areas near major public structures prevents people from carrying out their activities, which is detrimental to the local economy. In a city, the presence of a parking lot is also strategic in terms of security: a dark and abandoned place would prove much safer if it were used as a lighted parking area [2]. The parking sector in Europe and North America is going through an innovation process towards intelligent systems. In addition to adopting advanced automation and

software solutions for parking reservations and payments, it is gradually integrating with other mobility solutions, also thanks to the use of communications and information technology. These systems use detection devices to determine the occupation status of the parking lot. The detection devices are different: cameras, gates at the parking entrance, or sensors normally installed on the road surface. The accuracy of the algorithms and the usability of the applications are the determining factors for the success of the intelligent parking system [3,4].

Parking automation cannot be limited to parking management and the payment process. Parking lots are often crowded, with hundreds of cars frantically looking for a place to leave their car in order to reach their destination as quickly as possible. This activity can lead to dangerous situations for pedestrians and cars; in fact, accidents that occur inside parking lots are frequent, especially for underground ones where visibility is not excellent. Even if the speed of cars is limited, the accidents that occur can create serious damage to people and cars, triggering further possible conflicts between the people involved. For example, we can think of the parking lot of a shopping center where families go to buy necessities. Often, parents are accompanied by children who can get out of control because of the carts with food that they have to push towards the car. Such a situation in a scenario characterized by numerous moving cars can really become dramatic. Furthermore, underground car parks in particular are isolated and often dark places, where acts of violence against users can easily occur. In large cities, episodes of violence perpetrated against women are frequent in the parking lots which, at certain hours of the day, become almost deserted. In these cases, it is easy for an attacker to attack people, aware of the fact that someone is unlikely to run to the rescue of the victim. Being underground, these environments do not have GSM (Global System for Mobile Communications, London, UK) signal coverage and therefore it becomes impossible to request an intervention by the police forces if you have time to do so [5,6]. To address these security concerns, car parks are often equipped with video surveillance systems that monitor parking areas. Parking lots represent a demanding challenge for the video surveillance technology used: they have low ceilings and problematic lighting ratios which combine very dark areas with very bright areas. As a rule, car parks are certainly not inviting places; however, even here, it is necessary to guarantee the safety of users. The structure of the car parks and the need to contain hardware costs can create shadow cones that make control in some areas more difficult. Video monitoring is left to the parking staff, who must simultaneously monitor multiple monitors in sequence. This activity is characterized by a high percentage of inaccuracies due to the low image quality. Furthermore, the operator's ability to focus attention on the emergency scenario is a subjective factor that depends on multiple variables, not guaranteeing a measurable result in advance [7,8]. To improve the performance of a video surveillance system, an automatic image recognition system can be adopted to identify an emergency. A problem arises in the definition of an emergency scenario: images with nearby cars, or images with people in the vicinity of cars, or images with groups of people do not generally identify an emergency scenario. This limit can be overcome by integrating an acoustic monitoring system of the parking areas in a video surveillance system. The hardware upgrade has a modest cost, requiring only microphones, which are already provided in some areas—for example, in automatic cash machines or at the entrances and exits of parking lots. By using auxiliary microphones and an automatic audio event recognition system, it will be possible to significantly improve the security system of an indoor car park. Furthermore, the location of the sound source could guide an automatic rotation system of the video camera, which, in this way, could cover a larger area than the fixed ones [9,10]. The sound emission associated with the parking of a car can be divided into several phases, which are generally the path of the access roads to the parking lanes, the search for a parking spot, the actual parking operation, and the opening and closing of the door. The type of vehicle used also emits different noise levels. Added to these are the noises emitted by other sources, such as the sound of shopping carts in shopping centers, pedestrian steps, music from cars and piped music systems, and people who chat or talk on the phone. All these noises vary according to the parking areas and the periods of the day. In fact, as with other sources, the noise of the underground car park also depends on the time of day. Some car parks are subject to intense

traffic only at a specific time of day; for example, the parking areas near the business centers will be more busy in the morning and in the evening, when people go to the office and then return home. On the contrary, in entertainment places such as restaurants, bars, and hotels, there will be more cars after working hours. Identifying a sound source is not an easy task because of the variance in the sounds. In recent times, various authors have proposed methods for the automatic identification of sounds using algorithms based on machine learning [11,12].

In this work, a procedure is developed to automatically identify sound events in an underground garage. The sounds of a parking sector are detected with the use of sound sensors. These sounds are analyzed by a sound detector based on convolutional neural networks. The study focused on the automatic identification of car crashes.

The rest of the paper is organized as follows: in Section 2, the materials and methods used are described in detail; first, the tools used for the records and the techniques used are presented, and we then move on to analyze the extraction techniques of the characteristics used and finally explore convolutional neural networks. In Section 3, the results obtained in this study are described, providing an adequate discussion of the results achieved. In Section 4, conclusions are provided, with some possible examples of use in real life and the possible evolution of the research.

## 2. Methodology

The aim of this work is to develop a procedure to detect the presence of a car crash in a complex acoustic scenario such as an underground car park. The problem of identifying audio events in an underground car park is highlighted for reasons related to security risk mitigation. These environments are often theaters of aggression or more simply of accidents in which the need for immediate intervention by the rescue operators becomes crucial [13–15]. The method involves the use of acoustic sensors that detect the acoustic signals of the parking areas in real time. The detected acoustic signal is used as input by a system based on the use of convolutional neural networks that identify the occurrence of a car crash. In the event of detection, the attention of the surveillance operator is focused by activating an alert that displays the cameras of the video surveillance system of that area, and, in the case of emergency acknowledgement, it can notify the emergency services. The developed methodology is presented through a flowchart with indications of all the phases in Figure 1.

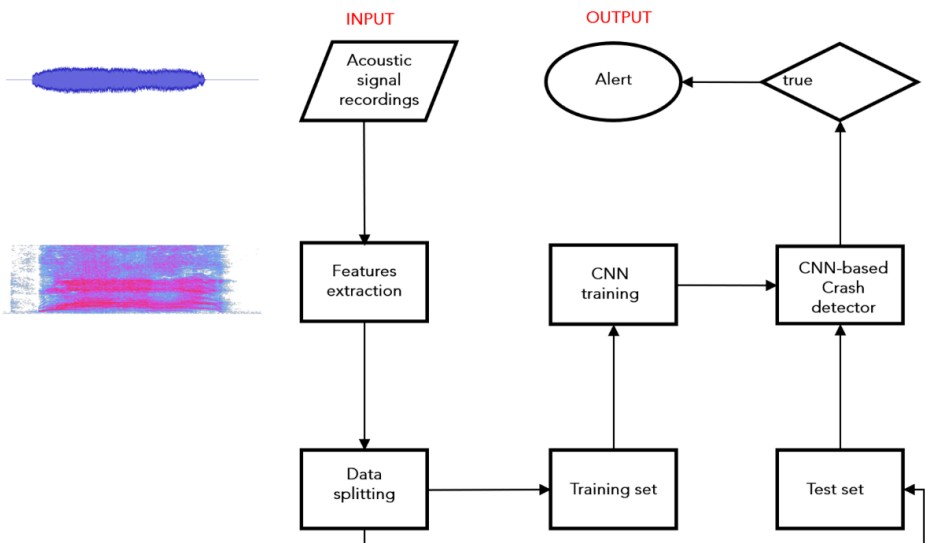

**Figure 1.** Flow chart of the automatic procedure for detecting a crash car in an underground car park. The steps provided by the procedure are the real-time recording of signals with acoustic sensors, extraction of features, training of a model based on convolutional neural networks, and development of a CNN (convolutional neural network) based crash car detector. If a car crash is detected, then an alert is raised which activates the parking surveillance operator.

## 2.1. Acoustical Signal Recordings

Several sessions of recordings of the typical sounds of an underground car park were made. An underground car park for a shopping center on the outskirts of a large city was chosen as the location. It is a large area divided into compartmental areas for fire prevention, in which the designer has sought the best and most efficient general arrangement of parking spaces and lanes according to the size of each section. The driveways and surface public roads have been designed to ensure maximum efficiency, and the directions of the pedestrian paths have been designed to ensure access to the shopping areas in the shortest possible time [16].

The critical points were highlighted during peak hours: in such periods, given the large number of vehicles present, the visibility of the maneuvering lanes is reduced, with an increase in possible crash cases between a car exiting the parking stall and a car in transit in the maneuvering aisle. Furthermore, the criticality of the areas of access to the compartment sections for the cars coming from the parking access ramps was highlighted; these cars risk colliding with the cars coming from the maneuvering lanes. For these reasons, it was decided to use these stations for sound recording [17].

The recordings were made with a high-quality Zoom H1 Handy Recorder with X-Y microphone. The recorder was placed on a tripod at a height of around 2.5 m from the floor. The recorder was placed in a position in the compartment area which guaranteed sufficient coverage of all parking stalls. Various scenarios were measured: cars coming from the access ramps, cars passing through the maneuvering lanes, cars maneuvering to occupy a parking stall, cars maneuvering to leave a parking stall, and cars parked with the engine running.

Each recording was divided into sections lasting around 60 s in order to obtain an adequate number of samples to be used in the training and testing phase. In this way, around 300 samples were extracted, equally distributed between the two identification classes (NoCrash, Crash). The samples labeled Crash were subsequently processed: to simulate a car accident, the typical sounds of a low-speed car accident were added to the recordings made in situ.

## 2.2. Signal Descriptor Selection

The audio signals recorded were then analyzed to extract features capable of identifying the emergency event. This is a crucial procedure in the event identification process, as it is thanks to these descriptors that the classifier will be able to highlight the anomalies as they will represent its input [18].

We have already listed some of the characteristic sounds of an underground parking area, to which are added all the noises of the air treatment systems. It is a complex acoustic scenario, with signals characterized by acoustic spectra with contributions in a wide range of frequencies with different levels. It follows that a time domain analysis would be inadequate for classifying events. It then becomes necessary to transfer to the frequency domain in order to extract the energy levels in the different frequencies. The Fourier operator allows us to switch from the time domain to the frequency domain through a projection of the signal acquired over time on an orthonormal basis of complex exponentials. When we apply the Fourier transform, we do nothing but switch between two different representations of the same signal: the original signal is in the time domain, while the transformed signal is in the frequency domain [19,20]. The fast Fourier transform (FFT) is a mathematical transformation to a function, f, represented by Equation (1):

$$f : R^n \rightarrow C \tag{1}$$

which matches an *F* function, represented by the Equation (2):

$$F(\xi) = \frac{1}{(2\pi)^{\frac{n}{2}}} \int_{R^n} e^{-i\xi t} f(x) dx \tag{2}$$

where

- *f(x)* is a real function of the real variable *t*.

- $F(\xi)$ is the Fourier transform of *f(x)*.

An important feature of this transformation is that its computational cost is extremely low. In fact, to carry out the basic change of a vector of n components, around 3/2 n *$\log_2$n operations are enough if n is an integer power of 2. Furthermore, the transformation is well conditioned and the fast algorithms for its calculation are numerically stable. For these particularly favorable characteristics, the discrete Fourier transform finds numerous uses in different fields of mathematics and its applications [21].

Once the FFT of the signals has been calculated, a spectrogram of the recorded sounds is extracted. A spectrogram is the representation of a sound using a Cartesian in which time is represented on the abscissa, the frequency on the ordinate. In addition, the frequency content is always represented with a color map. Therefore, in the spectrogram, the signal is represented with three variables: frequency, time, and intensity (color scale). The color map is built according to simple rules: dark colors represent low intensity sound, and light colors represent high intensity [22,23].

### 2.3. Convolutional Neural Network

The convolutional neural network (CNN) is one of the most common deep learning algorithms. It is used for the processing of data characterized by a particular grid topology: a CNN is able to emphasize local relationships, starting from adjacency structures present in the data, through automatic and adaptive learning of patterns from low to high level [24]. CNNs therefore represent the main model used in the field of computer vision and in general in applications that require object recognition and artificial vision. It is an architecture inspired by the biological structure of the visual cortex, in which there is a hierarchy of two basic types of cells: simple and complex cells. Simple cells react to primitive patterns present in sub-regions of the visual field, called receptive fields, while complex cells synthesize the information from the former to identify more complex structures. Similarly, the neurons present in a convolutional layer are connected to sub-regions of the previous level and are not affected by the signals located outside that area. The receptive fields can also overlap: the neurons of a CNN therefore produce spatially correlated results [25].

It is therefore possible to identify the main difference with a fully connected neural network (FFNN): while in a fully connected neural network, the number of parameters to be learned increases with increasing input size, a convolutional neural network reduces the number of parameters thanks to the reduced number of connections, shared weights, and sub-sampling. A convolutional neural network can have dozens or hundreds of layers, each of which is made up of different filters used to detect different features and build the corresponding feature maps. In the case of images, in fact, filters are applied to each input (called kernels) and, through a convolution operation, feature maps are generated. The latter will be used as input for the next layer. The filters of the first layers look for very simple features—for example, edges—to take on more and more complex shapes, able to uniquely define the object [26].

The architecture of a convolutional neural network includes several blocks, such as convolutional layers, pooling layers, and fully connected layers. The architecture of the convolutional neural network can be divided into two parts: feature detection, which then deals with the extraction of features through operations such as convolution, pooling, and ReLU (rectified linear unit), and classification, which generates the predicted output through the use of fully-connected and softmax layers (Figure 2).

The optimization of the network parameters takes place in the same way as described previously: the function signal is propagated forward in the network up to the final layer, where the gradient of the loss function is computed and retro-propagated to allow the updating of the weights through a gradient descent algorithm [27].

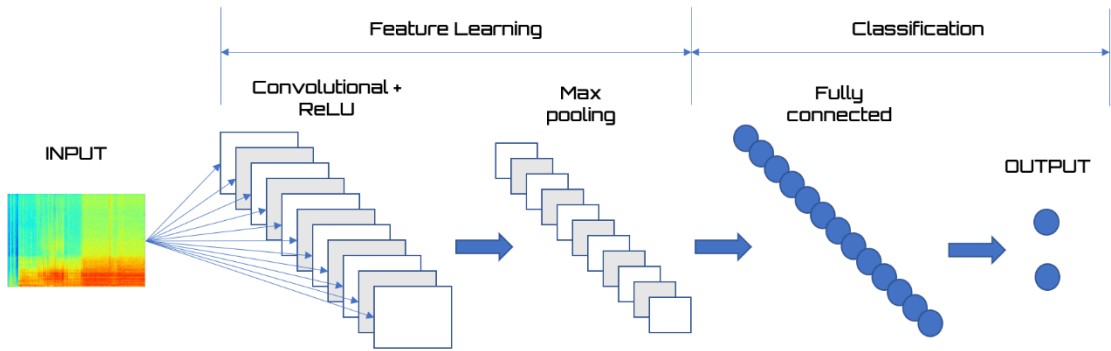

**Figure 2.** Example of architecture of a convolutional neural network for binary classification.

### 2.3.1. Convolutional Layer

In the convolution, a filter, called a kernel, is applied to a multidimensional array of numbers, called the input tensor. This filter represents a mask of multidimensional parameters limited in height and width but which extends over the entire depth of the input volume. The kernel analyzes the input tensor in height and width: in each position, the scalar product between the kernel and the covered portion of input is calculated in order to obtain the output value in the current position. The output volume obtained at the end of the process is called the activation map or the functionality map [28].

### 2.3.2. ReLU Layer

The convolution layer is typically followed by a non-linear activation function: only the activated features are passed to the next layer. The most used function is the rectified linear unit (ReLU), which performs a thresholding operation on each element by mapping the negative values to zero and keeping the positive values. It has been demonstrated that the use of the ReLU function allows faster and more effective training than traditional units such as the hyperbolic tangent [29].

### 2.3.3. Pooling Layer

The pooling layer performs a non-linear sub-sampling operation that reduces the transversal size of the activation maps, leaving their depth unchanged, with the aim not only of simplifying the output of the previous convolutional layer but also of introducing invariance translational to small displacements and distortions and, consequently, making it more robust than the localization of the features. Resuming the analogy with the visual cortex, the pooling unit was inspired by the behavior of complex cells: by capturing a growing visual field, they are able to learn spatial hierarchies of feature patterns, resulting in less sensitivity to slight displacements in the position of the salient features. The most popular form of the pooling operation is max pooling, which performs non-linear sub-sampling by dividing the input into rectangular regions and returning the maximum value within each window. It is a layer that does not have appreciable weights but for which a set of hyper-parameters must be set before training, such as filter size, pitch (stride), and padding. If the size of the filter, r, and stride are equal, the operation reduces the size of the input tensor for each channel by a factor, r [30].

### 2.3.4. Fully Connected Layer (FC)

The activation maps output from the last feature extraction block are transformed into a vector and connected to one or more fully connected layers, in which each value of the input vector is connected to all the values of the next layer. The last fully connected layer will generate a vector of size, K, equal to the number of classes in which the input must be classified [31].

2.3.5. Softmax Layer

The activation function applied to the last fully connected layer generally differs from those used in the previous levels and will be selected based on the task required from the network. In the multi-class learning paradigm, the softmax function is used, which normalizes the K real values obtained from the last FC layer in probability of belonging to the K classes in question [32].

## 3. Results and Discussion

In this study, a procedure for automatically identifying crashes between cars in an underground garage is developed. To start, the sounds of a sector of a parking lot were recorded with the use of a microphone. The microphone was positioned to cover the entire compartment sector; in addition, the recording sessions were performed during peak periods in which the car park was mostly occupied by customers' cars.

### 3.1. Processing of Recorded Signals

The recordings covered several acoustic scenarios representative of the sounds that are produced inside an underground car park. The sounds produced by people moving to and from cars were recorded: noise of heels on the pavement, drag of shopping carts, conversation between people, screams of children. Subsequently, the noises produced by the cars were recorded in the common maneuvering operations: closing of the car doors, maneuver for the occupation of the parking stall, movement of the cars in the lanes in search of a free stall, passage of the cars on the access ramps to the compartment sector. Finally, noises from the air circulation system were recorded. Furthermore, recordings of combinations of these noises were made in order to better represent the real situation. A total of 150 audio tracks of around 1-min duration were recorded. To simulate a car crash, the audio tracks were processed by adding a typical sound from a low speed car crash. Figure 3 shows the results of this processing.

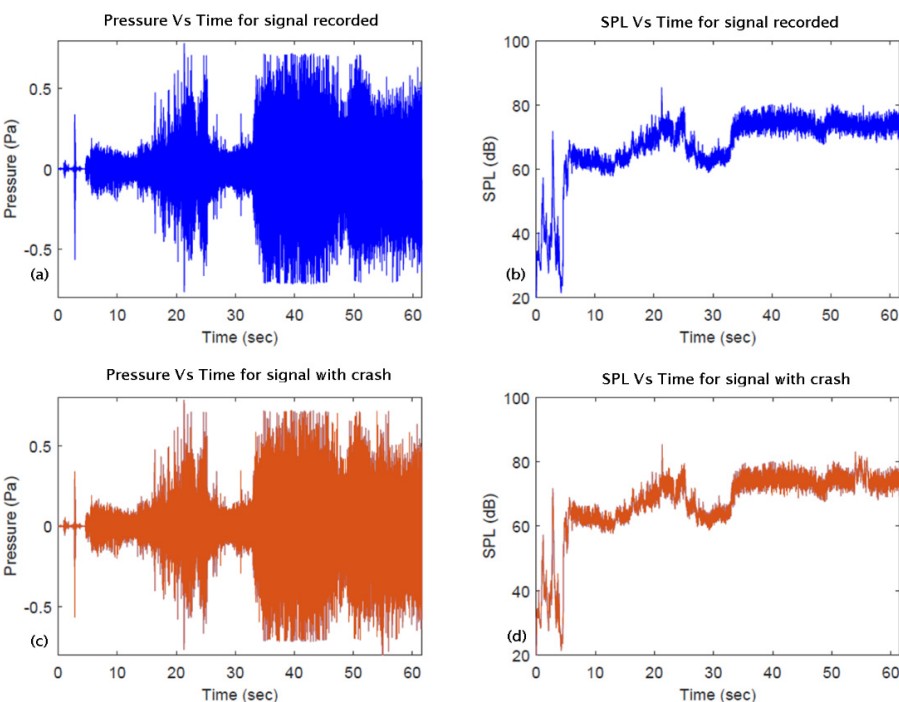

**Figure 3.** Example of processing the recorded signal. (**a**,**b**), in blue, at the top, refers to a signal recorded in the underground car park where there is the noise of a car maneuvering when leaving the parking stall. (**c**,**d**), in red, below the crash signal, has been added to the same signal.

Figure 3 shows the results of the processing carried out on a recorded audio signal of just over a minute. The graph shows the trend of sound pressure (left) and the trend of sound pressure level (SPL). SPL is the pressure level of a sound, measured in decibels (dB). It is obtained through the following equation:

$$SPL = 20 * log_{10}\left(\frac{p}{p_0}\right) \tag{3}$$

where

- $p$ is the root mean square of the pressure level.
- $p_0$ is a reference value for sound pressure, which, in air, assumes the standard value of 20 μPa.

In Figure 3a, a recorded audio track is shown in blue, relating to the maneuvering procedures exiting a parking stall. At the bottom of the same figure (Figure 3b) is shown in red the same audio track processed with the addition of the characteristic sound of a car accident. It is possible to notice a slight difference between the two tracks in the middle of the 50–60 s range. This difference is minimal and therefore represents a real challenge for the classification of the event. The difficulty in identifying the event using a representation of the signal over time demonstrates the limits of this procedure, which cannot be used. It is necessary to move on to the analysis of the signals in the frequency domain.

### 3.2. Feature Extraction

The first method used for the frequency analysis of stationary signals uses a bank of bandpass filters, which is a series of devices which each allow only a certain frequency range to pass, leaving out the components of the sound at higher and lower frequencies. By connecting a measuring instrument to the output of each filter, it is possible to measure the signal level that belongs to the frequency range. In a graph, a bandpass filter can be represented, with a zone in which the gain is almost constant and equal to 0 db, and with two zones, on either side of the first one, in which the gain decreases to negligible values. It should be noted that an ideal filter should have a gain curve like a rectangular pulse, but if the device is made of passive components, the rising and falling edges can never be vertical. Figure 4 shows the average spectra of two signals in bands of one-third octaves.

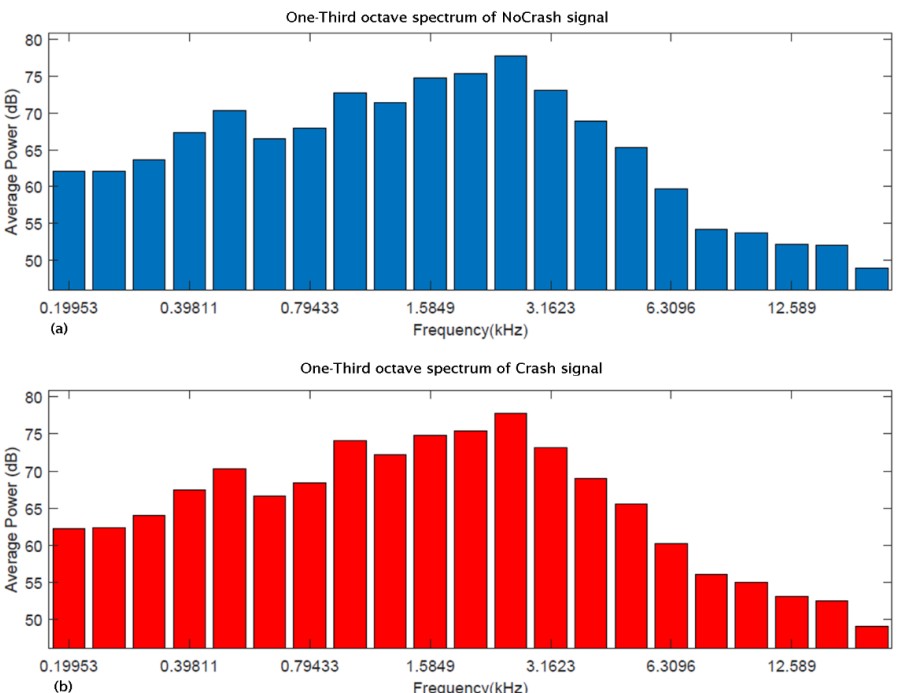

**Figure 4.** One-third octave band spectrum: (**a**) at the top in blue bars refer to a signal recorded in the underground car park where there is the noise of a car maneuvering when leaving the parking stall. (**b**) in red, at the bottom, the crash signal has been added to the same signal.

In frequency analysis, the bands are defined to have a constant width or a width that is proportional to the lower frequency of the band. The central frequency of the band is linked to its extreme frequencies, higher frequency, and lower frequency, through the geometric mean. The bands used in acoustics, present in many technical standards and regulations, are established according to international conventions. A band subdivision with constant percentage amplitude is the octave band, widely used in music. It is defined as a doubling in frequency, so having the frequency axis scaled by octaves means having a bank of filters with a constant percentage height, so that each successive band is twice the width of the previous one. An octave band can be divided into the corresponding three bands in thirds of an octave.

From the analysis of Figure 4, there are no major differences between the two diagrams. There are no frequency bands capable of discriminating between the two signals—that is, such as to allow the classification of the signal. Altogether, this indicates that this descriptor is not adequate for the identification of the crash event. We next investigated what would happen if we traced the spectrograms of the two signals. As previously anticipated, a spectrogram is the representation of a sound using a Cartesian in which time is represented on the abscissa, the frequency on the ordinate. In addition, the frequency content is always represented with a color map. Figure 5 shows the spectrograms of the two signals.

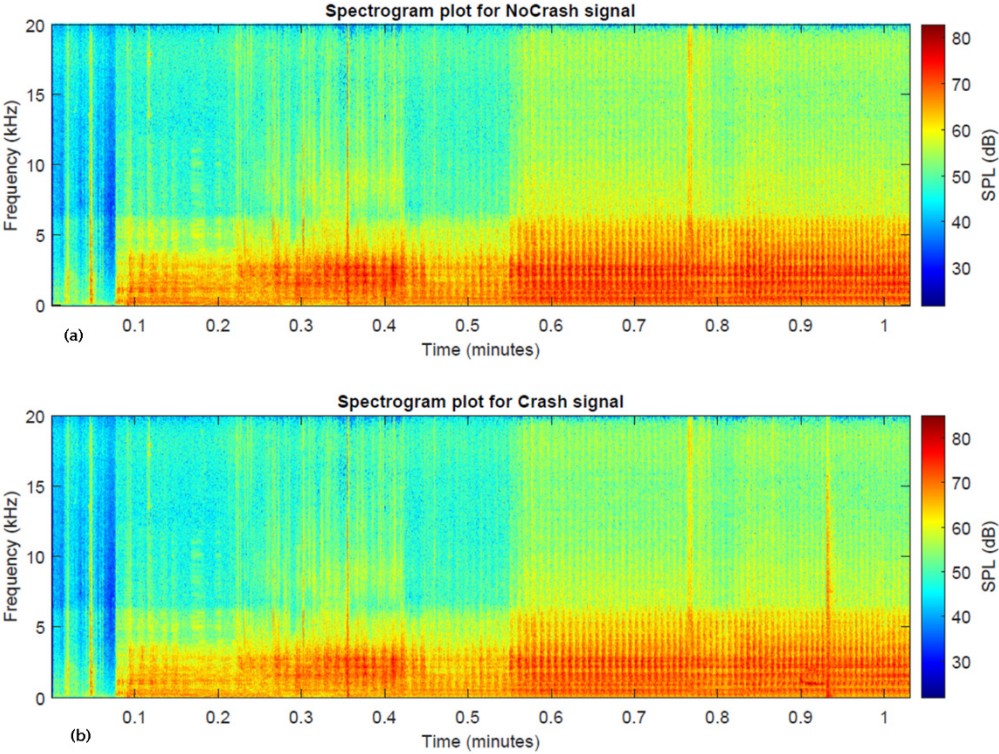

**Figure 5.** Spectrograms of the two signals: (**a**) at the top refers to a signal recorded in the underground car park where there is the noise of a car maneuvering when leaving the parking stall. (**b**) at the bottom, the crash signal has been added to the same signal.

From the analysis of Figure 5, it is possible to notice a broadband contribution towards the final part of the signal. This is the distinctive feature of the crash between cars that we have added to the signal recorded in the underground car park. This tells us that the spectrogram can represent a valid descriptor of the event that we are trying to identify: a visual analysis of the event is not able to solve the problem. To automate the event identification procedure, it is necessary to use a technology that fully exploits the characteristics of an image. Convolutional neural networks are particularly suitable for identifying objects in the image; for this reason, we decided to use this technology in the procedure.

The images obtained were subsequently processed by carrying out simple transformations without changing their structure. The images were subject to random rotation and flipping. In this way, we were able to increase the number of images to be used in the subsequent training and test phases from 300 images to 1200.

### 3.3. Sound Event Classification Using Convolutional Neural Network

The recorded sounds and those processed with the addition of the crash event were divided into two sets: training set and test set. The classification model was trained using the training data, while its performance was assessed using the test set. The proportion of confidential data for training and testing was at the analyst's discretion. The accuracy of the classifier was assessed based on the accuracy achieved by the classifier on the test data. For our purposes, we divided the 1200 sounds into a training set equal to 70% of the available sounds (840 sounds), equally distributed between NoCrash events and Crash events. The remaining 30% equal to 360 sounds, also equally distributed between NoCrash events and Crash events, were used to test the model's performance. For each sound, a spectrogram was developed, which was then saved as a png image with $800 \times 800$ pixels.

A model based on convolutional neural networks for the identification of the crash event in a complex acoustic scenario has been developed. The model is based on an architecture with three hidden levels, each of which is composed in sequence of a convolutional layer, a pooling layer, and a ReLU layer. Subsequently, a flatten layer was inserted in order to reduce the map obtained from one-dimensional information, and a fully connected layer was inserted, to then close with a densely connected NN layer with a softmax activation function that returned the probability of belonging to the two classes according to which the data have been labeled (NoCrash, Crash). All the layers of the elaborated model are shown in Table 1.

**Table 1.** Convolutional neural network-based model architecture.

| Layer Type | Description | Shape |
|---|---|---|
| Input | Spectrogram image ($800 \times 800$) png format | ($800 \times 800 \times 3$) |
| 1° Hidden | 2D spatial convolution for images<br>Max pooling operation for 2D spatial data<br>ReLu activation function | ($399 \times 399 \times 32$)<br>($199 \times 199 \times 32$)<br>($199 \times 199 \times 32$) |
| 2° Hidden | 2D spatial convolution for images<br>Max pooling operation for 2D spatial data<br>ReLu activation function | ($199 \times 199 \times 64$)<br>($99 \times 99 \times 64$)<br>($99 \times 99 \times 64$) |
| 3° Hidden | 2D spatial convolution for images<br>Max pooling operation for 2D spatial data<br>ReLu activation function | ($99 \times 99 \times 64$)<br>($49 \times 49 \times 64$)<br>($49 \times 49 \times 64$) |
| Flatten | Dimensionality reduction using a flatten operation<br>Random deactivation of some neurons via dropout | (153,664)<br>(153,664) |
| Fully connected | Layer of neurons interconnected with each other<br>ReLu activation function<br>Random deactivation of some neurons via dropout | (64)<br>(64)<br>(64) |
| Output | Densely- Layer of neurons interconnected with each other<br>Softmax activation function | (2)<br>(2) |

In the architecture shown in Table 1, starting from the input data, each subsequent layer processes the information provided in input and sends it to the next layer. In this way, feature maps are extracted that allow us to locate the characteristic information of that audio event. These feature maps will then be used to classify events, thus returning increasingly complex processing. After the training phase performed by the hidden convolutional layers, the classification phase is performed by a network

densely connected with a softmax activation function that returns the probability of belonging to its outgoing classes.

In the first phase of the procedure shown in Figure 1, training of the model is foreseen using 70% of the available data. Subsequently, the remaining 30% of the data is used for the model test; in this way, the validation of the algorithm takes place with the use of new data, never processed by the model until that moment. It is important that the set of tests is independent of the set used for sampling in order to avoid too optimistic estimates. For the evaluation of the model's performance, the accuracy in the classification of the two sound classes (NoCrash, Crash) was calculated. Accuracy returns the percentage of correct classifications on all observations submitted to the model. The model developed using convolutional neural networks provided an accuracy of 0.87, showing the strength of the procedure for identifying a crash in an underground car park. The accuracy of a prediction indicates how close the expected value of a quantity is to the real value of that quantity. In our case, the real value is available as we have measured the accuracy of the model on test data that have been properly labeled. A classification model is a mathematical function that uniquely determines the class to which a statistical unit belongs, based on the values observed for the variables of interest. Its predictive accuracy depends on the ability to correctly classify new units, regardless of the class they come from. A result such as that obtained (0.87) tells us that the model can correctly classify 87 cases out of 100 that have been presented to it. Furthermore, it should be noted that this performance was obtained on a sample equally representing the two classes.

The accuracy returned by the CNN-based model proved to be in line with the results obtained from previous studies that used pattern recognition in different areas. Bardou et al. [33] used CNN to classify lung sounds. The authors extracted the functionality of the local binary model (LBP) from the visual representation of the audio files using spectrograms: the accuracy returned by the model was 0.80. Salamon et al. [34] used CNN to classify ambient sound. The accuracy returned by the model was 0.75–0.80, and the performance was improved by performing a data-augmentation operation. Piczak [35] used CNN to classify short sound clips of ambient sounds. A deep model consisting of two convolutional levels with max pooling and two fully connected levels was trained on a low-level representation of audio data using segmented spectrograms. The results of the model returned accuracy ranging from 0.65 to 0.80 depending on the dataset used to confirm the importance of the data in the correct classification of the audio sources.

## 4. Conclusions

In the social life of people, mobility is a crucial element to guarantee meeting and cultural and service exchange. Among the means of transport, the car is most widely used by populations living on the outskirts of large cities or in medium and small cities. For motorists, a prerequisite for reaching a destination is the availability of parking in the immediate vicinity. This is the reason why large car parks are built near large shopping and service centers to ensure that users can park. Often, these car parks are very large and arranged on several underground floors. In such environments, checking security is difficult and requires a large amount of resources. For some time now, large underground car parks have been equipped with video surveillance systems which, in some cases, have sophisticated systems for detecting moving objects [36]. Although these systems have often been studied to improve performance [37,38], they currently do not guarantee adequate control due to the difficulty in identifying an emergency.

Machine learning-based algorithms have recently been widely used in various fields both for regression problems [39,40] and for classification problems [41–44]. In this work, a procedure was developed to automatically identify sound events in an underground garage. The sounds of a parking sector were detected with the use of microphones. These sounds were subsequently processed to add a crash event and then labeled according to one of two classes (NoCrash, Crash). For the classification of events, a model based on convolutional neural networks was developed.

From the experimental results, the following conclusions can be drawn:

1.  The characterization of the crash noise between cars did not highlight any trends in the time domain, meaning that an analysis in this domain is not able to identify the event.
2.  The comparison between the spectra in the frequency domain in the one-third octave band during the two scenarios (NoCrash, Crash) shows that the two signals are comparable and no tonal components are highlighted. This confirms that the ambient noise in such scenarios is so complex that it is not possible to distinguish between the different acoustic sources, even using this descriptor.
3.  The comparison between the spectrograms of the two scenarios demonstrated a broadband component at the event. This indicates that the spectrogram is a descriptor capable of discriminating between the two scenarios.
4.  A CNN-based rating system has proven to be able to identify the occurrence of a crash between cars with an accuracy of 0.87, demonstrating the strength of the procedure for identifying an accident in an underground parking garage.

This procedure can be used to enrich modern indoor video surveillance systems by simply adding a microphone to the cameras. A CNN-based system will then be able to identify an audio event and issue an alert that will focus the surveillance operator's attention on the parking sector. The procedure used in this study can be extended to other sound sources in order to identify specific sounds in emergency situations that require an emergency response, in which the identification of possible risks for users becomes difficult with traditional technologies.

The limits shown by CNN relate to computational costs, which are high, since it is an image processing process; however, this is a disadvantage rather than a limit. This problem can be overcome with better processing hardware that takes advantage of the graphics processing units (GPU). In addition, the support of a surveillance operator is required in order to verify the alert signal; therefore, the procedure is not fully automated. The latter limit can be overcome by integrating the recognition of the images detected by the video cameras into the proposed classification system.

**Funding:** This research received no external funding.

**Conflicts of Interest:** The author declares no conflict of interest.

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
