# Peer review of "Sound Event Detection in Underground Parking Garage Using Convolutional Neural Network"

_2504-2289, doi:10.3390/bdcc4030020_

Round 1

Reviewer 1 Report

This paper presented a detection algorithm for sound events in an underground parking garage. The paper is not very well written and authors should do the proof reading of this paper from third party specially the English language experts. My main concern in this work are the followings:

1) Why the authors used the convolutional neural network on the binary classification of sound events where many techniques have already proposed and used, e.g. decision trees, random forests, Bayesian networks, support vector machines, logistic regression, and Probit model. More specifically, authors should evaluate the advantages of using CNNs over others.

2) Which are the benefits of classifying sound events from an underground parking garage compared to segmenting objects in video surveillance except for cost down. From my point of view, it is reasonable to argue that modern cameras provide the possibility of applying algorithms on the edge, thus avoiding the need of transferring frames from all scenarios to apply computer vision algorithms in an expensive common server.

3) I cannot find where and how the dataset built? Also, what’s the distribution for each category. Please, explain and discuss these issues in the manuscript.

4) I don’t see any experiments in the manuscript. The authors must provide comparisons with more recent and top-performing techniques. The authors must analyse this to provide insights for future research.

Author Response

This paper presented a detection algorithm for sound events in an underground parking garage. The paper is not very well written and authors should do the proof reading of this paper from third party specially the English language experts.

AUTHOR RESPONSE: The author thanks the reviewer for his careful work on the paper. The proposed comments have certainly enriched this work with content, focusing the author's attention on some aspects not adequately addressed. I dealt with all the comments and I modified the paper accepting the proposed suggestions.

My main concern in this work are the followings:

1) Why the authors used the convolutional neural network on the binary classification of sound events where many techniques have already proposed and used, e.g. decision trees, random forests, Bayesian networks, support vector machines, logistic regression, and Probit model. More specifically, authors should evaluate the advantages of using CNNs over others.

AUTHOR RESPONSE: The sounds of an urban scenario are complex signals in which identifying a source is not a simple task. Convolutional neural networks (CNN) are widely used for the processing of data characterized by a particular grid topology. A CNN is able to emphasize local relationships starting from adjacency structures present in the data, through automatic and adaptive learning of patterns from low to high level. This is the reason why CNNs represent the main model used in the field of computer vision and in general in applications that require object recognition and artificial vision. In this work, sound spectrograms were created by returning images that CNNs are able to analyze and recognize patterns identifying dangerous situations.

The algorithms mentioned by the reviewer have demonstrated excellent results in binary classification. In fact, the author has used these algorithms in other works that I indicate below:

  • Iannace, G., Ciaburro, G., & Trematerra, A. (2019). Wind turbine noise prediction using random forest regression. Machines, 7 (4), 69.
  • Iannace, G., Ciaburro, G., & Trematerra, A. (2020). Acoustical unmanned aerial vehicle detection in indoor scenarios using logistic regression model. Building Acoustics, 1351010X20917856.

The difficulty in identifying sources in a complex sound scenario requires the identification of suitable descriptors that are able to discriminate between the different sources. This represents a preliminary phase that must be performed before using these algorithms. In the case of convolutional networks, the filters that extract the features are contained in the model and this phase is incorporated into the model.

2) Which are the benefits of classifying sound events from an underground parking garage compared to segmenting objects in video surveillance except for cost down. From my point of view, it is reasonable to argue that modern cameras provide the possibility of applying algorithms on the edge, thus avoiding the need of transferring frames from all scenarios to apply computer vision algorithms in an expensive common server.

AUTHOR RESPONSE: Sound is an important sense in our life and that is why we also use it to orient ourselves and to identify possible threats. My work is aimed at developing a methodology to improve the security systems of underground parking. As a case study, I considered identifying a crash between cars. But later developments in the work involve identifying a person's request for help from an attacker. These situations are characterized by specific sound events that can be classified by a system based on artificial intelligence. Segmenting objects is a technology that gives excellent performance. But the information contained in sounds represents a resource that a modern surveillance system cannot neglect.

3) I cannot find where and how the dataset built? Also, what’s the distribution for each category. Please, explain and discuss these issues in the manuscript.

AUTHOR RESPONSE: I added additional descriptions to the data collection process. The information is contained in section 2.1, in section 3.2, and in section 3.3. Anyway, I extracted that information below:

Each recording was divided into pieces lasting about 60 s to obtain an adequate number of samples to be used in the training and testing phase. In this way, about 300 samples were extracted, equally distributed between the two identification classes (NoCrash, Crash). The samples labeled Crash were subsequently processed: To simulate a car accident, the typical sounds of a low-speed car accident have been added to the recordings made in situ.

The images obtained were subsequently processed by carrying out simple transformations without changing their structure. The images were subject to random rotation, and flipping. In this way we were able to increase the number of images to be used in the subsequent training and test phases, from 300 images to 1200.

For our purposes we have divided the 1200 sounds into a training set equal to 70% of the available sounds (840 sounds) equally distributed between NoCrash events and Crash events. The remaining 30% equal to 360 sounds also equally distributed between NoCrash events and Crash events will be used to test the model's performance. For each sound, a spectrogram was developed which was then saved as a png image with 800x800 pixels.

4) I don’t see any experiments in the manuscript. The authors must provide comparisons with more recent and top-performing techniques. The authors must analyse this to provide insights for future research.

AUTHOR RESPONSE: I proceeded to make a comparison between the performance of the algorithm used in this paper with those obtained in several studies that used similar technology.

Reviewer 2 Report

This is a very thorough research study that explains the sound event detection in underground parking garage using convolutional neural network. I read the manuscript with great interest and believe its topic is important and relevant. The authors performed a careful and thorough review of the literature, as the section was very informative and substantial. Appropriate theoretical framework was applied. I found the methodological part to be well justified and reasonable for this type of analysis. Although the manuscript is overall well-written and structured, it might benefit from additional spell/language checking. However, I have some comments which I would like to be addressed before the acceptance of this paper.

Major comments

  1. Modify your abstract following the MDPI’s guidelines. The abstract should be a total of about 200 words maximum. The abstract should be a single paragraph and should follow the style of structured abstracts, but without headings: 1) Background: Place the question addressed in a broad context and highlight the purpose of the study; 2) Methods: Describe briefly the main methods or treatments applied. Include any relevant preregistration numbers, and species and strains of any animals used. 3) Results: Summarize the article's main findings; and 4) Conclusion: Indicate the main conclusions or interpretations. The abstract should be an objective representation of the article: it must not contain results which are not presented and substantiated in the main text and should not exaggerate the main conclusions.
  2. What was the key novelty of the present study?
  3. What was the motivation behind using the convolutional neural network?
  4. Please make comparison of your study findings with the past studies.
  5. What are the limitations of your study?

I look forward to read the final version.

Author Response

This is a very thorough research study that explains the sound event detection in underground parking garage using convolutional neural network. I read the manuscript with great interest and believe its topic is important and relevant. The authors performed a careful and thorough review of the literature, as the section was very informative and substantial. Appropriate theoretical framework was applied. I found the methodological part to be well justified and reasonable for this type of analysis. Although the manuscript is overall well-written and structured, it might benefit from additional spell/language checking. However, I have some comments which I would like to be addressed before the acceptance of this paper.

AUTHOR RESPONSE:

The author thanks the reviewer for his careful work on the paper. The proposed comments have certainly enriched this work with content, focusing the author's attention on some aspects not adequately addressed. I dealt with all the comments and I modified the paper accepting the proposed suggestions.

Major comments

Modify your abstract following the MDPI’s guidelines. The abstract should be a total of about 200 words maximum. The abstract should be a single paragraph and should follow the style of structured abstracts, but without headings: 1) Background: Place the question addressed in a broad context and highlight the purpose of the study; 2) Methods: Describe briefly the main methods or treatments applied. Include any relevant preregistration numbers, and species and strains of any animals used. 3) Results: Summarize the article's main findings; and 4) Conclusion: Indicate the main conclusions or interpretations. The abstract should be an objective representation of the article: it must not contain results which are not presented and substantiated in the main text and should not exaggerate the main conclusions.

AUTHOR RESPONSE:

As suggested, I have rewritten the abstract according to the MDPI's guidelines. I have eliminated the superfluous information and I have focused the attention on the purposes of this work, briefly shown the results obtained.

What was the key novelty of the present study?

AUTHOR RESPONSE:

The novelty of this study is the use of sound signals for monitoring activities in an underground car park. Modern security systems use video surveillance systems in which activities are monitored based on the images of the cameras. In this technology sounds are added to images, to add additional information. The system then uses artificial intelligence to automatically recognize dangerous situations. Sounds identifying dangerous situations are detected to activate an automatic alert that draws the attention of surveillance in that area.

What was the motivation behind using the convolutional neural network?

AUTHOR RESPONSE:

The sounds of an urban scenario are complex signals in which identifying a source is not a simple task. Convolutional neural networks (CNN) are widely used for the processing of data characterized by a particular grid topology. A CNN is able to emphasize local relationships starting from adjacency structures present in the data, through automatic and adaptive learning of patterns from low to high level. This is the reason why CNNs represent the main model used in the field of computer vision and in general in applications that require object recognition and artificial vision. In this work, sound spectrograms were created by returning images that CNNs are able to analyze and recognize patterns identifying dangerous situations.

Please make comparison of your study findings with the past studies.

AUTHOR RESPONSE:

I proceeded to compare the results of my study with those of other researchers who have adopted convolutional networks for pattern recognition. I have also added a detailed description of the importance of the described and accuracy for evaluating the performance of a forecasting model.

What are the limitations of your study?

AUTHOR RESPONSE:

The limits shown by CNN relate to computational costs which are expensive since it is an image processing process. But this is more a disadvantage than a limit. This problem can be overcome with better processing hardware that takes advantage of the Graphics processing units (GPU). In addition, the support of a surveillance operator is required to verify the alert signal, therefore the procedure is not fully automated. The latter limit can be overcome by integrating the recognition of the images detected by the video cameras into the proposed classification system.

Reviewer 3 Report

1. Please enhance the introduction highlighting the motivation, problem statement, state of the approaches, and the contributions. Some of the content is irrelevant.
2. Please validate the proposed method in terms of hardware resources and computationally efficiency of the algorithm in comparison to the existing methods.
3. The main contribution of the paper is unclear. Please highlight the significance of the work in comparison to existing approaches in this field.
4. Conclusions do not reflect the contributions. The outcome of the analysis and the importance of the work is not adequately brought out.
5. English in the paper should be revised thoroughly.
6. Many format errors in this paper should be revised.

Author Response

  1. Please enhance the introduction highlighting the motivation, problem statement, state of the approaches, and the contributions. Some of the content is irrelevant.

AUTHOR RESPONSE:

Thank you. I tried to improve the introduction by highlighting the motivation, the statement of the problem, the state of the approaches and the contributions. I removed some of the content that might have seemed irrelevant. The objective of the Introduction was this: First I highlighted the importance of parking for the economy, then I mentioned the automation of parking with the use of devices. Afterwards I talked about the safety of parking lots and the limits of video surveillance systems. So I introduced the use of audio signals as an integration of a video surveillance system, and finally I mentioned machine learning. I also added an outline of the paper at the end of the section.

  1. Please validate the proposed method in terms of hardware resources and computationally efficiency of the algorithm in comparison to the existing methods.

AUTHOR RESPONSE:

I proceeded to make a comparison between the computational efficiency of the algorithm used in this paper with those obtained in several studies that used similar technology. I also added a paragraph on the hardware resources needed for CNN training.

  1. The main contribution of the paper is unclear. Please highlight the significance of the work in comparison to existing approaches in this field.

AUTHOR RESPONSE:

Modern security systems use video surveillance systems in which activities are monitored based on the images of the cameras. In this technology sounds are added to images, to add additional information. The system then uses artificial intelligence to automatically recognize dangerous situations. Sounds identifying dangerous situations are detected to activate an automatic alert that draws the attention of surveillance in that area.

  1. Conclusions do not reflect the contributions. The outcome of the analysis and the importance of the work is not adequately brought out.

AUTHOR RESPONSE:

To highlight the importance of the work, I have listed the results obtained in the conclusions section. I also added possible additional uses to identify emergency situations from analyzing audio recordings.

  1. English in the paper should be revised thoroughly.

AUTHOR RESPONSE:

I have made a complete revision of the English for the whole paper.

  1. Many format errors in this paper should be revised.

AUTHOR RESPONSE:

I proceeded to carry out a complete revision for the entire paper for the correction of format errors.

Round 2

Reviewer 1 Report

I have read the authors’ feedback from the revision. However, I think authors did not address all of my comments. In fact, I noticed that the core part of this work, i.e., convolutional neural network, is highly repetitive with a conventional convolutional neural network, e.g., LeNet-5. However, the LeNet-5’s paper (Gradient-Based Learning Applied to Document Recognition) is not cited and discussed. Hence, the novelty of this work is ambiguous. Moreover, the description about the main algorithm is not self-contained in this paper.

It is unclear to me whether the superiority of the proposed method over other machine learning-based methods comes from the newly collected dataset, since it seems that the authors did not retrain other learning-based methods on the collected dataset for fair comparison.

Hence, I expected to see that the proposed CNN-based approach works well, and even, it outperforms the machine learning-based methods (e.g. decision trees, random forests, Bayesian networks, support vector machines, logistic regression, and Probit model.) to show the advantages of using CNNs over others. Otherwise I don’t see any particular reasons to use CNN for a simple binary classification problem. However, the authors still did not offer this comparison in the revision.

Although some results of this paper look good, the comparison and analysis are limited. There should be some metrics at least for comparison. For example, the comparison show in following paper: Pruning Fuzzy Neural Network Applied to the Construction of Expert Systems to Aid in the Diagnosis of the Treatment of Cryotherapy and Immunotherapy. Big Data Cogn. Comput. 2019, 3, 22.

In summary, the weakest point to me is the novelty and advantage of the proposed method. It's not clear that the proposed method can fully address the problem or is consistently better than others. I’m also not convinced due to the lack of thorough comparison and analysis. In this respect, I’m leaning toward negative.

Author Response

I have read the authors’ feedback from the revision. However, I think authors did not address all of my comments. In fact, I noticed that the core part of this work, i.e., convolutional neural network, is highly repetitive with a conventional convolutional neural network, e.g., LeNet-5. However, the LeNet-5’s paper (Gradient-Based Learning Applied to Document Recognition) is not cited and discussed. Hence, the novelty of this work is ambiguous. Moreover, the description about the main algorithm is not self-contained in this paper.

AUTHOR RESPONSE: I thank the reviewer again for his contribution to making this work more readable. I dealt with all the comments proposed by the auditor. This work aims to provide a new methodology for the recognition of audio events in complex sound scenarios. In this context, convolutional neural networks represent a valid and universally recognized tool for the classification of signals. The article by LeCun has made school and I apologize for my forgetfulness that I proceeded to repair by inserting the article in the references. In my view, the novelty of my work lies in the use of CNN for the identification of audio events in a complex sound scenario.

It is unclear to me whether the superiority of the proposed method over other machine learning-based methods comes from the newly collected dataset, since it seems that the authors did not retrain other learning-based methods on the collected dataset for fair comparison.

AUTHOR RESPONSE: In this work I have used CNN to identify audio events in a complex sound scenario. The difficulty of identifying the sources in a complex sound scenario requires the identification of suitable descriptors capable of discriminating between the different sources. This represents a preliminary phase that must be performed before using these algorithms. In the case of convolutional networks, the filters that extract the functionalities are contained in the model and this phase is incorporated in the model. Furthermore, as CNNs are widely used in computer vision, this methodology could be integrated into an automatic event recognition system based on images and sounds. In this case the CNN verrebebro used both for the images of the cameras and for the sounds detected by microphones. This integration was proposed as a study development.

Hence, I expected to see that the proposed CNN-based approach works well, and even, it outperforms the machine learning-based methods (e.g. decision trees, random forests, Bayesian networks, support vector machines, logistic regression, and Probit model.) to show the advantages of using CNNs over others. Otherwise I don’t see any particular reasons to use CNN for a simple binary classification problem. However, the authors still did not offer this comparison in the revision.

AUTHOR RESPONSE: In order to use algorithms based on decision trees, random forests, Bayesian networks, support vector machines, logistic regression and Probit model it is necessary to extract features capable of discriminating audio events. Each sound has different characteristics that can be identified by different descriptors that influence the model's performance. In this study, the spectrograms of the sounds were extracted, obtaining images which are subsequently classified by CNN. The results obtained from the CNN-based model have been compared with other papers available in the literature that have used CNNs.

Although some results of this paper look good, the comparison and analysis are limited. There should be some metrics at least for comparison. For example, the comparison show in following paper: Pruning Fuzzy Neural Network Applied to the Construction of Expert Systems to Aid in the Diagnosis of the Treatment of Cryotherapy and Immunotherapy. Big Data Cogn. Comput. 2019, 3, 22.

AUTHOR RESPONSE: I thank the reviewer for the in-depth suggestion: I found the article very interesting, so much so that I included it in the references. In my article I focus on the methodology for identifying sound events in support of an underground parking surveillance system.

In summary, the weakest point to me is the novelty and advantage of the proposed method. It's not clear that the proposed method can fully address the problem or is consistently better than others. I’m also not convinced due to the lack of thorough comparison and analysis. In this respect, I’m leaning toward negative.

AUTHOR RESPONSE: The novelty lies in the use of the methodology in a context so far unexplored. Video surveillance does not consider audio events which instead contain crucial information for the automatic identification of emergency situations. The advantage of the proposed method was confirmed by the accuracy returned by the model. The usefulness of using CNNs is provided by the possibility of integrating audio recognition into an automatic image recognition system, thus using the same technology.

Reviewer 2 Report

All my comments are addressed hence, manuscript is accepted. 

Author Response

The author thanks the reviewer for his careful work on the paper. The proposed comments have certainly enriched this work with content, focusing the author's attention on some aspects not adequately addressed.

Reviewer 3 Report

The comments are listed in the attached pdf.

Author Response

The author thanks the reviewer for his careful work on the paper. The proposed comments have certainly enriched this work with content, focusing the author's attention on some aspects not adequately addressed. I have attached the pdf with the replies to the comments

Round 3

Reviewer 1 Report

Authors have addressed most of my comments in this revision. The background of this paper is somehow related to computer vision, especially video surveillance systems. However, there is no survey on some recent video surveillance systems to support the background in this paper. For example, low-rank and sparse matrix decomposition based surveillance (Robust PCA via Outlier Pursuit. IEEE Transactions on Information Theory, 58(5), 3047-3064; A Robust Moving Object Detection in Multi-Scenario Big Data for Video Surveillance. IEEE Trans. Circuits Syst. Video Techn. 29(4): 982-995 (2019) and the variant with non-negativity constraint based surveillance (MahNMF: Manhattan Non-negative Matrix Factorization. arXiv:1207.3438v1.). I will recommend accepting this paper after taking this problem.

Author Response

I thank the reviewer again for the suggestions. I proceeded to insert the references in the paper.

Reviewer 3 Report

Please revise carefully according to my comments.

Author Response

I thank the reviewer again for the suggestions. I proceeded to analyze the suggestions contained in revision 2. I proceeded to add changes to those already made previously. I have not separated Section 2 Materials and Methods because I had already taken steps to insert subsections that define the different sections. I hope the changes made are sufficient.